

# The Importance of Considering Sub-grid Cloud Variability When Using Satellite Observations to Evaluate the Cloud and Precipitation Simulations in Climate Models

Hua Song[1], Zhibo Zhang[1, 2*], Po-Lun Ma[3], Steven Ghan[3], and Minghuai Wang[4]

1. Joint Center for Earth Systems Technology, UMBC, Baltimore, MD
2. Physics Department, UMBC, Baltimore, MD
3. Atmospheric Sciences and Global Change Division, Pacific Northwest National Laboratory, Richland, WA
4. Institute for Climate and Global Change Research & School of Atmospheric Sciences, Nanjing University, Nanjing, China

Corresponding Author:
   Dr. Zhibo Zhang
   Email: Zhibo.Zhang@umbc.edu
   Phone: +1 (410) 455 6315



## Abstract

Satellite cloud observations have become an indispensable tool for evaluating the general

circulation models (GCMs). To facilitate the satellite and GCM comparisons, the CFMIP (Cloud
Feedback Model Inter-comparison Project) Observation Simulator Package (COSP) has been
developed and is now increasingly used in GCM evaluations. In this study, we use COSP cloud
simulations from the Super-Parameterized Community Atmosphere Model (SPCAM5) and
satellite observations from the Moderate Resolution Imaging Spectroradiometer (MODIS) and
CloudSat to demonstrate the importance of considering the sub-grid variability of cloud and
precipitation when using the COSP to evaluate GCM simulations. We carry out two sensitivity
tests: SPCAM5 COSP and SPCAM5-Homogeneous COSP. In the SPCAM5 COSP run, the sub-
grid cloud and precipitation properties from the embedded cloud resolving model (CRM) of
SPCAM5 are used to drive the COSP simulation, while in the SPCAM5-Homogeneous COSP
run only grid mean cloud and precipitation properties (i.e., no sub-grid variations) are given to
the COSP. We find that the warm rain signatures in the SPCAM5 COSP run agree with the
MODIS and CloudSat observations quite well. In contrast, the SPCAM5-Homogeneous COSP
run which ignores the sub-grid cloud variations, substantially overestimates the radar reflectivity
and probability of precipitation compared to the satellite observations, as well as the results from
the SPCAM5 COSP run. The significant differences between the two COSP runs demonstrate
that it is important to take into account the sub-grid variations of cloud and precipitation when
using COSP to evaluate the GCM to avoid confusing and misleading results.


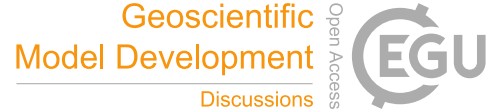

## 1. Introduction

Marine boundary layer (MBL) cloud, as a strong modulator of the radiative energy budget of the Earth-Atmosphere system, is a major source of uncertainty in future climate change projections of the general circulation models (GCM) (Cess et al., 1996; Bony and Dufresne, 2005). Improving MBL cloud simulations in the GCMs is one of the top priorities of the climate modeling community. As the cloud parameterization schemes in the GCMs become increasingly sophisticated, there is a strong need for comprehensive global satellite cloud observations for model evaluation and improvement. However, the fundamental definitions of clouds in GCMs differ dramatically from those used for satellite remote sensing, which hampers the use of satellite products for model evaluation. In order to overcome this obstacle, the Cloud Feedback Model Inter-comparison Project (CFMIP) community has developed an integrated satellite simulator, the CFMIP Observation Simulator Package (COSP) (Zhang et al., 2010; Bodas-Salcedo et al., 2011). COSP has greatly facilitated and promoted the use of satellite data in the climate modeling community to expose and diagnose issues in GCM cloud simulations (e.g., Marchand et al., 2009; Zhang et al., 2010; Kay et al., 2012; Pincus et al., 2012; Kay et al., 2016; Song et al., 2017).

Warm rain is a unique and important feature of MBL clouds. It plays an important role in determining the macro- and micro-physical properties of MBL clouds, in particular, the cloud water budget (e.g., Stevens et al., 2005; Wood, 2005; Comstock et al., 2005). Many previous studies have investigated the warm rain simulation in GCMs using the COSP simulators. These studies reveal a common problem in the latest generation of GCMs, i.e., the drizzle in MBL clouds is too frequent in the GCM compared with satellite observations (e.g., Zhang et al. 2010; Franklin et al. 2013; Suzuki et al. 2015; Takahashi et al., 2017; Jing et al., 2017; Song et al.,



2017, Bodas-Salcedo et al. 2008; Stephens et al. 2010; Bodas-Salcedo et al. 2011; Nam and
Quaas 2012; Franklin et al. 2013; Jing et al., 2017). One possible reason for the excessive warm
rain production in GCMs could be the model's inaccurate representation of physical processes,
such as auto-conversion and accretion that govern the precipitation efficiency in warm MBL
clouds. Due to the lack of sub-grid variability of microphysical quantities in most large-scale
models, the auto-conversion parameterization is overly aggressive so that the models tend to
produce precipitation too quickly (Lebsock et al. 2012, 2013, Song et al. 2017).

The radar observations of warm rain from CloudSat and collocated MODIS (Moderate

Resolution Imaging Spectroradiometer) cloud observations are extremely useful data for
assessing and improving the GCM simulations of MBL clouds and their precipitation process.
However, the dramatic spatial resolution differences between the conventional GCM (~100km)
and satellite observations (~1km) become a challenging obstacle for the satellite and GCM
comparisons. To overcome this obstacle, the COSP first divides the grid-level cloud and
precipitation properties (e.g., grid-mean cloud water and rain water) into the so-called "sub-
columns" that are conceptually similar to "pixel" in satellite observation. Then for each sub-
column the COSP satellite-simulators (e.g., COSP-CloudSat and COSP-MODIS) simulate the
satellite measurements (e.g., radar reflectivity) and retrievals (e.g., MODIS cloud optical depth
and effective radius) which become directly comparable with satellite data. Ideally, the sub-
column generation in COSP should be consistent with the sub-grid cloud parameterization
scheme in the host GCM. However, in practice sub-grid variations of cloud and precipitation are
often ignored or treated crudely in the COSP simulation for a number of possible reasons. First
of all, the COSP is an independent package and it takes substantial efforts to implement in the
COSP a sub-grid cloud generation scheme that is consistent with the host GCM. Secondly, a

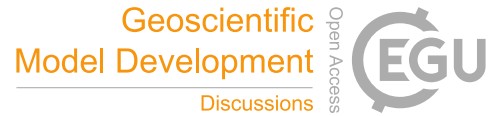

simple sub-column generation scheme helps alleviate the computational cost associated with the
COSP simulation. Last but certainly not least, the users of the COSP might not be fully aware of
the consequences of ignoring the sub-grid cloud and precipitation variability in the COSP
simulations.

The current version (v1.4) of COSP provides a built-in highly simplified sub-column

generator. It accounts only for the sub-grid variability of the types of hydrometeors and ignores
the variability of mass and microphysics within each hydrometeor type.  The water content and
microphysical properties (i.e., droplet effective radius and optical thickness) of each hydrometeor
are horizontally homogenous among all the sub-columns that are labeled as the same type (i.e.,
stratiform or convective).  Here we refer to the current scheme as the "*homogenous hydrometeor*
*scheme*".  The uncertainties and potential biases caused by the *homogenous hydrometeor scheme*
can be significant and should not be overlooked. A simple hypothetical example is provided in
Figure 1 to illustrate the importance of accounting for the sub-grid variability of rainwater in
simulating the CloudSat radar reflectivity. To be consistent with the two-moment cloud
microphysics scheme (Morrison and Gettelman, 2008) that is widely used in the GCMs, we
assume the sub-grid distribution of rainwater to follow the exponential distribution.  In this
example, the grid-mean rainwater mixing ratio ($\bar{q}$) is set to be 0.03 g/kg (dashed blue line in
Figure 1a). Using the Quickbeam simulator (Haynes et al., 2007) in COSP, we simulated the
corresponding 94-GHz CloudSat radar reflectivity, which is shown in Figure 1b.  The grid-mean
radar reflectivity based on the exponentially distributed rainwater (i.e., with sub-grid variance) is
about 4 dBZ (solid red line in Figure 1b).  In contrast, if the sub-grid variation of rainwater is
ignored, the radar reflectivity corresponding to $\bar{q}$ = 0.03 g/kg is 13 dBZ (dashed blue line in
Figure 1b). The substantial difference between the two indicates that ignoring the sub-grid



variability of hydrometeors could cause significant overestimation of grid-mean radar reflectivity
simulation, which in turn could complicate and even mislead the evaluation of GCMs.

The objective of this study is to investigate and demonstrate to the GCM modeling

community the importance of considering the sub-grid variability of cloud and precipitation
properties when evaluating the GCM simulations using COSP. Here we employ the Super-
parameterized Community Atmosphere Model Version 5 (SPCAM5, Wang et al., 2015) to
provide the sub-grid cloud and precipitation hydrometeor fields for a comparison study of the
simulated radar reflectivity and warm rain frequencies by COSP. Fundamentally different from
the convectional cloud parameterization schemes in GCMs, SPCAM5 consists of a two-
dimensional cloud-resolving model (CRM) embedded into each grid of a conventional CAM5
(Khairoutdinov and Randall, 2003; Wang et al., 2015).   The sub-grid cloud dynamical and
microphysical processes are explicitly resolved at a 4-km resolution in SPCAM5. We carry out
two sensitivity tests: SPCAM5 COSP and SPCAM5-Homogeneous COSP. In the SPCAM5
COSP run, the sub-grid cloud and precipitation properties from the embedded CRMs of
SPCAM5 are used to drive the COSP simulation. In the SPCAM5-Homogeneous COSP run, the
default *homogenous hydrometeor scheme* of COSP mentioned above is used to generate the sub-
grid cloud and precipitation fields for the COSP simulation. The outputs from the two runs are
compared with the collocated CloudSat and MODIS observations to assess the potential
problems in both runs, and also to understand the impacts of omitting sub-grid cloud variations
in the COSP simulations.

The rest of the paper is organized as follows: Section 2 describes the model, COSP and

satellite data used in this study. Results are represented in Section 3. Finally, Section 4 provides
general conclusions and remarks.




## 2. Description of Model, COSP and Satellite Observations

### 2.1. Model

The model used in this study is SPCAM5, an application of the Multiscale Modeling Framework (MMF) (Randall et al., 2003; Khairoutdinov et al., 2005, 2008; Tao et al., 2009) to CAM5 (Neale et al., 2010), which uses the finite volume dynamical core at 1.9° latitude × 2.5° longitude resolution with 30 vertical levels and 600-s time step. The embedded 2-D CRM in each CAM5 grid cell includes 32 columns at 4 km horizontal grid spacing and 28 vertical layers coinciding with the lowest 28 CAM5 levels. The CRM runs with a 20-s time step. Details of the SPCAM5 can be found in Wang et al. (2011; 2015). The simulations are run in a "constrained meteorology" configuration (Ma et al., 2013; 2015) to facilitate model evaluation against observations, in which the model winds are nudged toward the Modern Era Reanalysis for Research Applications (MERRA) reanalysis with a relaxation timescale of 6 hours (Zhang et al., 2014). The SPCAM5 simulations are performed from September 2008 to December 2010 (28 months). The last 24 months (January 2009-December 2010) outputs of the simulations are used for analysis.

### 2.2. COSP

We used COSP Version 1.4, which has no scientific difference from the latest version COSP2 (Swales et al., 2018). Currently, COSP provides simulations of ISCCP (International Satellite Cloud Climatology Project), CALIPSO (Cloud-Aerosol Lidar and Infrared Pathfinder Satellite Observation), CloudSat, MODIS, and MISR (Multi-angle Imaging SpectroRadiometer) cloud measurements and/or retrievals (Bodas-Salcedo et al., 2011). In this study, we will focus on the MODIS and CloudSat simulators (Pincus et al., 2012; Haynes et al., 2007). COSP has



three major parts, each controlling a step of the pseudo-retrieval process: (1) the *sub-column*
*generator* of COSP first distributes the grid-mean cloud and precipitation properties from GCM
into the so-called sub-columns that are conceptually similar to "pixels" in satellite remote
sensing. (2) the *satellite simulators* simulate the direct measurements (e.g., CloudSat radar
reflectivity and CALIOP backscatter) and retrieval products (e.g., MODIS cloud optical
thickness and effective radius) for each sub-column using highly simplified radiative transfer and
retrieval schemes; (3) the *aggregation scheme* averages the sub-column simulations back to grid
level to obtain temporal-spatial averages that are comparable with aggregated satellite products
(e.g., MODIS level-3 gridded monthly mean products).

As mentioned in the Introduction, the COSP-v1.4 has a highly simplified built-in sub-

column generator based on the homogenous hydrometeor scheme. This scheme accounts only for
the sub-grid variability of the types of hydrometeors and ignores the variability of mass and
microphysics within each hydrometeor type. An example is provided in Figure 2 to illustrate
how this default sub-column generator of COSP-v1.4 distributes the grid-mean cloud and
precipitation into the sub-columns. We arbitrarily selected a grid (23°N and 150°E) with both
cloud and significant precipitation from our previous CAM5 simulations (Song et al., 2017).
Figure 2a shows the vertical profiles of the grid-mean total (stratiform plus convective) and
convective cloud fractions at the selected grid box. Figure 2b shows the vertical profiles of the
grid-mean mixing ratios of each type of hydrometeors. The sub-column generator of COSP takes
the grid-mean cloud fractions, hydrometeor mixing ratios and effective particle sizes (Figure 2a
and Figure 2b) as inputs to generate the sub-columns for the later satellite measurement and
retrieval simulation.





First, sub-columns (150 sub-columns generated in our example) are assigned as either
cloudy or clear at each model level by the Subgrid Cloud Overlap Profile Sampler (SCOPS),
which was developed originally as part of the ISCCP simulator (Klein and Jakob, 1999; Webb et
al., 2001). As illustrated in Figure 2c, the SCOPS assigns cloud to the sub-columns in a manner
consistent with the model's grid box average stratiform and convective cloud amounts (Figure
2a) and its cloud overlap assumption, i.e., maximum-random overlap in this case. The next step
is to determine which of the sub-columns generated by SCOPS contain precipitation
hydrometeors, e.g., rain and snow. This step is necessary and critical for the COSP CloudSat
radar simulator (Bodas-Salcedo et al., 2011) because radar reflectivity is highly sensitive to the
precipitation hydrometeors due to their large particle size (L'Ecuyer and Stephens, 2002; Tanelli
et al., 2008). The current sub-grid precipitation distribution scheme "SCOPS-PREC" is
developed and described in Zhang et al. (2010). Figure 2d shows the masking of precipitation
among the 150 sub-columns generated by the SCOPS-PREC for the example grid. After the
cloud and precipitation are masked, the last step is to specify the mass (i.e., mixing ratio) and
effective radius of hydrometeors for all the sub-columns occupied by clouds and/or precipitation.
The current scheme for this step is highly simplified. As shown in Figure 2e, it assumes the mass
and the microphysics of each type of hydrometeor to be horizontally homogeneous among all the
sub-columns that are occupied by this type of hydrometeor at a given model level. In other
words, at each model level the only difference among sub-columns is that they may be occupied
by different types of hydrometeors (Zhang et al., 2010).
In this study, we have carried out two COSP simulations using the 2-year SPCAM5
CRM outputs to investigate the importance of considering the sub-grid variations of cloud and
precipitation properties when evaluating the GCM simulations using COSP. The two COSP



simulations are marked as SPCAM5 COSP and SPCAM5-Homogeneous COSP, respectively.
For the SPCAM5 COSP simulation, we treat the sub-grid cloud and precipitation fields from the
CRM of SPCAM5 outputs as sub-columns of COSP without using the COSP sub-column
generator. For the SPCAM5-Homogeneous COSP simulation, we first average the sub-grid
cloud and precipitation fields (including both clear and cloudy sub-grids) from the CRM of
SPCAM5 to each CAM5 grid, and then input these grid-mean cloud and precipitation fields to
the default COSP-v1.4 sub-column simulator described above to generate the sub-column fields.
All the other processes of two COSP simulations are exactly same. The COSP simulator outputs
are produced from 6-hourly calculations and the number of sub-columns used here is 32. To
derive the probability of precipitation, we made some simple in-house modifications in COSP
v1.4 to write out the MODIS and CloudSat simulations for every sub-column. This allows us to
derive the joint statistics of COSP-MODIS and COSP-CloudSat simulations and compare them
with those derived from collocated MODIS and CloudSat level-2 products.
**2.3. Satellite Data**

The cloud measurements from the A-Train satellite sensors, namely MODIS and

CloudSat, are used for model-to-observation comparison. The newly released collection 6 (C6)
Aqua-MODIS cloud products (Platnick et al., 2017) are used to evaluate cloud fraction, cloud
optical thickness and cloud droplet effective radius.  For MBL cloud studies, CloudSat provides
valuable information on the warm rain process that cannot be achieved by a passive sensor like
MODIS.  The direct measurement of CloudSat is the vertical profile of 94-GHz radar reflectivity
by cloud and hydrometer particles (i.e., 2B-GEOPROF product), from which other information
such as vertical distribution of clouds and precipitation can be derived. The CloudSat 2B-
GEOPROF product (Marchand et al., 2008) is used for cloud vertical structure, radar reflectivity,





and identification of precipitation in MBL clouds. To prepare for the comparison of joint
statistics, we collocated 5 years (2006 ~ 2010) of pixel-level (i.e., level-2) MODIS and CloudSat
observations using the collocation scheme developed in Cho et al. (2008). Due to the low
sampling rate of CloudSat, we used 5 years (2006 ~ 2010) of observations, in comparison with
the 2-year model simulation (2009 ~ 2010), to obtain enough statistics. A sensitivity study
indicates that the inter-annual variability of MBL clouds is much smaller than the model-to-
observation differences.

In this study, we focus on the tropical and subtropical regions between 45°S and 45°N

(loosely referred to as "tropical and subtropical region"), where most stratocumulus and cumulus
regimes are found. We avoid high latitudes because satellite observations, namely MODIS, may
have large uncertainties to low solar zenith angles there (Kato and Marshak, 2009; Grosvenor
and Wood, 2014; Cho et al., 2015).

**3. Sensitivity Study: SPCAM5 COSP vs. SPCAM5-Homogeneous COSP**

First, we compare the Contoured Frequency by Altitude Diagram (CFAD) of tropical

clouds derived based on SPCAM5 COSP and SPCAM5-Homogeneous COSP simulations with
that derived from CloudSat 2B-GEOPROF product in Figure 3. The CFAD based CloudSat
observations displays a typical boomerang type shape that has been reported in many previous
studies (Bodas-Salcedo et al., 2011; Zhang et al., 2010; Marchand et al., 2009). Focusing on the
low clouds below 3km, we observe a rather broad distribution of radar reflectivity with a
maximum occurrence frequency around −30 dBZ ~ −20 dBZ followed by a long tail extending to
about 10 dBZ. As pointed out in previous studies, the peak around −30 dBZ ~ −20 dBZ is due to
non-precipitating MBL clouds and the precipitating clouds with increasing rain rate give rise to



the long tail. The CFAD based on two COSP simulations exhibits some characteristics similar to
the CloudSat observations, but also many noticeable differences. In particular, the two COSP
simulations both produce a much narrower range of radar reflectivity for low clouds, with
occurrence frequency clustered mostly around −25 dBZ in SPCAM5 COSP and around 0 dBZ in
SPCAM5-Homogeneous COSP. These results show that using the oversimplified COSP sub-
column generator (e.g., the homogeneous hydrometeor scheme) has non-negligible influences on
the simulated radar reflectivity and produces artificially high occurrences of large radar
reflectivity.

The systematic biases in simulated radar reflectivity by the COSP homogeneous

hydrometeor scheme might lead to the unjustified and biased evaluation of the warm rain
production in GCMs, since cloud column maximum radar reflectivity ($Z_{max}$) is often used to
distinguish precipitating from non-precipitating MBL clouds (Kubar and Hartmann, 2009;
Lebsock and Su, 2014; Haynes et al., 2009).

Next we compare the simulated and observed PDFs of $Z_{max}$ for all the sub-columns that

are marked as warm liquid clouds in the domain between 45°S and 45°N.  The warm liquid
clouds are defined by the cloud phase and cloud top pressure derived from the MODIS simulator
by the criteria that cloud phase is liquid and cloud top pressure is between 900 hPa and 500 hPa.
Big differences in the PDFs of $Z_{max}$ between the SPCAM5-Homogeneous COSP and the A-Train
observations, and between SPCAM5-Homogeneous COSP and SPCAM5 COSP are shown in
Figure 4.  First, in the A-Train observations, about 46% of warm liquid clouds detected by the
MODIS are not observed by the CloudSat.  These clouds are either too thin and therefore their
radar reflectivity is too weak to be detected by CloudSat, or they are too low and therefore suffer
the surface clutter issue (Marchand et al., 2008).  For those warm liquid clouds detected by both





the MODIS and CloudSat, the PDF of $Z_{max}$ peaks around -25 dBZ.  Second, in both COSP
simulations, almost all warm liquid clouds derived by the MODIS simulator have valid CloudSat
radar reflectivity larger than -40 dBZ. The PDFs of $Z_{max}$ in the SPCAM5 reasonably resemble
those in the A-Train observations.  However, significantly different from the other two, the
distribution of $Z_{max}$ in the SPCAM5-Homogeneous shifts to the large dBZ values and peaks
around 0 dBZ.  In previous studies (e.g., Takahashi et al., 2017), warm liquid clouds are
categorized to three different modes by $Z_{max:}$ non-precipitating mode ($Z_{max} < -15$ dBZ), drizzle
mode (-15 dBZ $< Z_{max} < 0$ dBZ) and rain mode ($Z_{max} > 0$ dBZ).  The simulated and observed
PDFs of $Z_{max}$ demonstrate that a large portion of warm liquid clouds is non-precipitating in the
observations and SPCAM5 COSP while most warm liquid clouds are precipitating (drizzle or
rain) clouds in the SPCAM5-Homogeneous COSP. The use of the COSP homogeneous
hydrometeor scheme gives us a dramatically different assessment of the warm rain production of
MBL clouds in the SPCAM5 model, i.e., if we consider the sub-column variability of cloud and
precipitation in the COSP simulation, we find that the SPCAM5 model can reproduce the
observed warm rain production quite well. However, if we ignore the CRM sub-grid variability
and use the homogeneous hydrometeor scheme, we may make the biased conclusion that the
SPCAM5 model performs badly in the simulation of warm rain production.

More significant differences between the SPCAM5 COSP and SPCAM5-Homogeneous

COSP simulations can be found from the spatial distributions of the probability of precipitation
(POP) in MBL warm clouds (Figure 5).  Here, the POP for a given grid box is defined as the
fraction of liquid-phase cloud identified by MODIS observations with $Z_{max}$ larger than a certain
threshold (i.e., $-15$ dBZ for drizzle or rain, 0 dBZ for rain, and 10 dBZ for heavy rain,
respectively) according to the collocated CloudSat observations with respect to the total



population liquid-phase clouds with the cloud top pressure between 500 hPa and 900 hPa in the
grid. Observations in Figure 5 suggest that roughly a third of MBL clouds observed by MODIS
in the tropical and subtropical region are likely precipitating (drizzle or rain), with a domain
averaged POP around 33%. The POP of drizzle plus rain has a distinct pattern: smaller (~15%)
in the coastal Sc regions and increasing to ~50% in the Cu cloud regions.  The observed POPs of
rain and heavy rain show similar spatial patterns as those of drizzle plus rain, with much smaller
domain averaged POP being about 12.5% and 3.3%, respectively.

In the same way as we define POP for observations, we define the POP for two COSP

simulations as the ratio of sub-columns that have COSP-CloudSat simulated $Z_{max}$ larger than a
certain threshold with respect to the total number of liquid-phase clouds identified by COSP-
MODIS.  As shown in Figure 5, two COSP simulations show dramatically different spatial
distributions of POPs.  The SPCAM5 COSP produces the similar POP patterns as those in the
observations, with the domain averaged POPs for drizzle or rain, rain and heavy rain being about
43%, 16% and 4.5%, respectively. However, the POPs in the SPCAM5-Homogeneous COSP are
substantially overestimated, with the domain averaged POPs for drizzle or rain, rain and heavy
rain being about 75%, 36% and 7%, respectively.  Using the COSP homogeneous hydrometeor
scheme will lead to the conclusion that the drizzle or rain is triggered too frequently (more than
double of the observations) in the SPCAM5 model, which obviously is not a fair assessment.

Previous studies find that the warm rain production in MBL clouds is tightly related to

the in-cloud microphysical properties of MBL clouds (e.g., Stevens et al., 2005; Wood, 2005;
Comstock et al., 2005).  Next, we check the dependence of POP on in-cloud properties liquid
water path (LWP) and on liquid cloud effective radius ($r_e$) in both observations and two COSP
simulations. Figure 6 shows the POPs of drizzle or rain (i.e., $Z_{max} > -15$ dBZ) as a function of in-





cloud LWP and $r_e$ overlaid by the joint PDF of LWP and $r_e$ (white contours) in the satellite
observations and two COSP simulations. The observed POPs of warm liquid clouds increase
monotonically with increasing in-cloud LWP and $r_e$, with high POPs concentrating on the
domain with large values of LWP and $r_e$ (i.e., LWP > 200 g/m$^2$ and $r_e$ > 15 μm). However, in the
two COSP simulations, especially the SPCAM5-Homogeneous COSP, at each joint bin the POPs
are much larger than those in the A-Train observations. When in-cloud LWP ($r_e$) is larger than
150 g/m$^2$ (17 μm), the dependence of POPs on in-cloud $r_e$ (LWP) is small. The joint PDFs of in-
cloud LWP and $r_e$ in the observations and two COSP simulations are also quite different. There
are more occurrences with large LWP and $r_e$ in the MODIS observations than the two COSP
simulations. The SPCAM5 COSP simulations have two peaks of the joint PDFs, which are
converted to one occurrence peak in the SPCAM5-Homogeneous COSP simulation by using the
COSP homogeneous hydrometeor scheme.
Based on the above comparisons, we can see that the oversimplified COSP sub-column
generator contributes to not only the narrow distribution of MBL cloud radar reflectivity, but
also to unrealistically high POPs in the SPCAM5 model. Besides, it also changes the distribution
of in-cloud microphysical properties, and the relationship between POPs and cloud
microphysical properties as well.

**4. Summary and Discussion**
This study presents a satellite-based evaluation of the warm rain production of MBL
cloud in the SPCAM5 model using two COSP simulations (SPCAM5 COSP and SPCAM5-
Homogeneous COSP), with the objective to demonstrate the importance of considering the sub-
grid variability of cloud and precipitation when using COSP to evaluate GCM simulations.





Through the SPCAM5 COSP simulations, in which the sub-column variability of cloud and
precipitation is considered, we find that the SPCAM5 model can reproduce the observed warm
rain production quite well. However, in the SPCAM5-Homogeneous COSP simulation, in which
we ignore the CRM sub-grid variability and use the COSP homogeneous hydrometeor scheme,
the simulated radar reflectivity and POPs in the SPCAM5 are significantly overestimated
compared to the observations.  Therefore, use of the COSP homogeneous hydrometeor scheme
gives us a significantly different assessment of warm rain production of MBL clouds in the
SPCAM5 model.

The systematic and significant biases due to the limitation of current homogeneous

hydrometeor scheme can mislead the evaluation of GCMs and should not be overlooked. In this
regard, an improved sub-column generator needs to be developed for COSP to account for the
sub-grid variances of cloud and/or hydrometer mass and microphysics.  A recent study of
Hillman et al. (2017) investigated the sensitivities of simulated satellite retrievals to subgrid-
scale overlap and condensate heterogeneity, and demonstrated the systematic biases in the
simulated MODIS cloud fraction and CloudSat radar reflectivity due to the oversimplified COSP
sub-column generator.  Their study also proposed a new scheme to replace the COSP current
sub-column generator, and showed that the new scheme can produce much better satellite
retrievals.  Implementing their sub-column heterogeneous hydrometeor scheme in COSP may
improve the GCM COSP simulations and give a better-justified assessment of the GCM
performance in simulating warm rain processes and cloud microphysical properties.

On the other hand, since the assumptions of sub-grid variability of cloud and

hydrometeors in different GCMs may be quite different, one universal sub-column hydrometeor
scheme may be not applicable to all models. Based on this consideration, the latest version



COSP version 2 enhances flexibility by allowing for model-specific representation of sub-grid
scale cloudiness and hydrometeor condensates and encourages the users to implement the same
sub-grid scheme as the host GCM for consistency (Swales et al., 2018). Nevertheless, our study
also suggests that any evaluation study of warm rain production in GCMs by using COSP
simulators should take this issue into account.



**Code and Data Availability:**

Details of SPCAM5 can be found in Wang et al. (2011). The host GCM in SPCAM5 is

the Community Atmospheric Model, Version 5 (see details on the CESM website at
http://www.cesm.ucar.edu/models/cesm1.1/cam/).   SPCAM5 has recently been merged with
CESM1.1.1    and    released    to    the    public    (Randall    et    al., 2013; https://svn-ccsm-
release.cgd.ucar.edu/model_development_releases/spcam2_0-cesm1_1_1). Codes of COSP V1.4
can be found in the website at https://github.com/CFMIP/COSPv1.  We used the collection 6 (C6)
Aqua-MODIS cloud products (Platnick et al., 2017), which can be downloaded from the NASA
website    at    https://ladsweb.modaps.eosdis.nasa.gov/api/v1/productPage/product=MYD06_L2.
The CloudSat data are distributed by the CloudSat Data Processing Center. The CloudSat 2B-
GEOPROF    product    we    used    is    downloaded    from    the    website    at
http://www.cloudsat.cira.colostate.edu/data-products/level-2b/2b-geoprof?term=42.







**Acknowledgements.** This research is supported by Department of Energy (DOE), Office of
Science, Biological and Environmental Research, Regional & Global Climate Modeling Program
(grant #DE-SC0014641). The Pacific Northwest National Laboratory is operated for the DOE by
Battelle Memorial Institute under contract DE-AC05-76RLO 1830. Minghuai Wang was
supported by the Minister of Science and Technology of China (2017YFA0604001). The
computations in this study were performed at the UMBC High Performance Computing Facility
(HPCF). The facility is supported by the U.S. National Science Foundation through the MRI
program (grant nos. CNS-0821258 and CNS-1228778) and the SCREMS program (grant no.
DMS-0821311), with substantial support from UMBC. The MODIS cloud products used in this
study are downloaded from NASA Level-1 and Atmosphere Archive and Distribution System
(https://ladsweb.modaps.eosdis.nasa.gov/). The CloudSat products are provided by CloudSat
Data Processing Center (http://www.cloudsat.cira.colostate.edu/).














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



**List of Figures:**


Figure 1. a) PDF of rainwater mixing ratio for rainwater when the horizontal variability of
rainwater is assumed to follow the exponential distribution. The vertical dashed blue line
indicates the mean value of rainwater mixing ratio as 0.03 g/kg. b) The corresponding PDF
of the CloudSat radar reflectivity simulated by COSP assuming the Marshall and Palmer
particle size distribution. The dashed blue line corresponds to the radar reflectivity based on
the mean rainwater 0.03 g/kg, and the solid red line corresponds to the grid-mean radar
reflectivity based on the PDF of rainwater mixing ratio.
Figure 2. a) The grid mean total (stratiform plus convective) and convective cloud fraction. b) the
grid mean mixing ratios of cloud and precipitation hydrometeors (LS_CLIQ: large-scale (i.e.,
stratiform) cloud water; LS_CICE: large-scale cloud ice; LS_RAIN: large-scale rain;
LS_SNOW: large-scale snow; LS_GRPL: large-scale graupel; CV_CLIQ: convective cloud
water; CV_CICE: convective cloud ice; CV_RAIN: convective rain; CV_SNOW: convective
snow). c) the distribution of large-scale and convective cloud among the sub-columns
generated by the SCOPS scheme (i.e., frac_out from scops.f). d) the distribution of large-
scale and convective precipitation among the sub-columns generated by the SCOPS-PREC
scheme (i.e., prec_frac from prec_scops.f). e) the mixing ratio (left panels) and effective
radius (right panels) of three hydrometeor types among the sub-columns.
Figure 3. Tropical averaged radar reflectivity-height histogram in the CloudSat observation (top),
the SPCAM5 CloudSat simulation (bottom left) and the SPCAM5_Homogeneous CloudSat
simulation (bottom right).





Figure 4. The histograms of column maximum radar reflectivity for liquid clouds over oceanic

regions from 45°S to 45°N in A-Train satellite observations, SPCAM5 COSP and SPCAM5-

Homogeneous COSP simulations.

Figure 5. Probability of precipitation (POP) of liquid clouds between 500hPa and 900hPa levels

in the satellite observations (left panel), the SPCAM5 COSP simulation (middle panel) and

the SPCAM5-Homogeneous COSP simulation (right panel). Three categories of precipitation:

drizzle plus rain (column $Z_{max}$ > -15 dBZ, top panels), rain (column $Z_{max}$ > 0 dBZ, middle

panels), and strong rain only (column $Z_{max}$ > 10 dBZ, bottom panels). Unit of POP is %.

Figure 6. POP (drizzle or rain) of liquid clouds at each LWP and liquid cloud effective radius in

the satellite observations (top), the SPCAM5 COSP simulation (bottom left) and the

SPCAM5-Homogeneous COSP simulation (bottom right). The white solid contours are joint

PDF of LWP and liquid cloud effective radius.  Units of POP and PDF are %.

















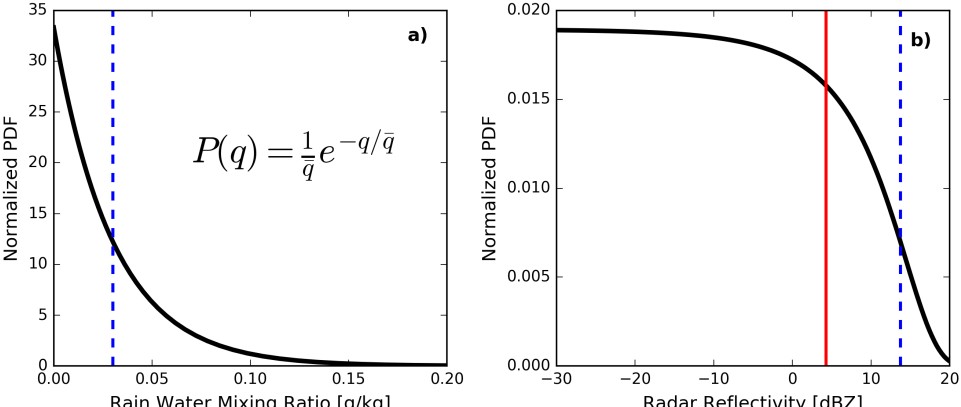



Figure 1. a) PDF of rainwater mixing ratio for rainwater when the horizontal variability of rainwater is assumed to
follow the exponential distribution. The vertical dashed blue line indicates the mean value of rainwater mixing ratio
as 0.03 g/kg. b) The corresponding PDF of the CloudSat radar reflectivity simulated by COSP assuming the
Marshall and Palmer particle size distribution. The dashed blue line corresponds to the radar reflectivity based on
the mean rainwater 0.03 g/kg, and the solid red line corresponds to the grid-mean radar reflectivity based on the PDF
of rainwater mixing ratio.












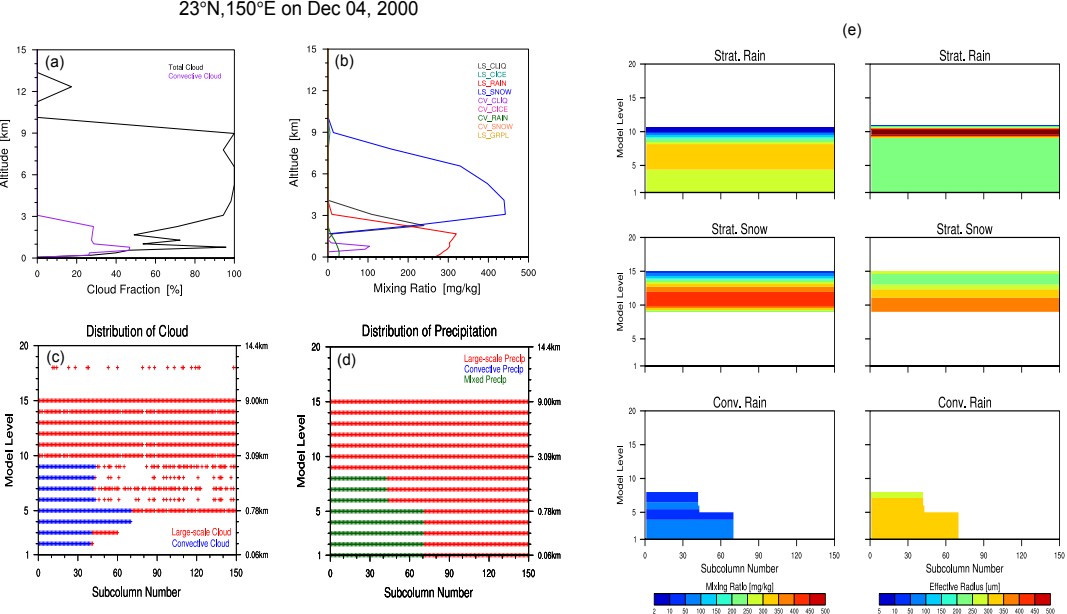



Figure 2. a) The grid mean total (stratiform plus convective) and convective cloud fraction. b) the
grid mean mixing ratios of cloud and precipitation hydrometeors (LS_CLIQ: large-scale (i.e.,
stratiform) cloud water; LS_CICE: large-scale cloud ice; LS_RAIN: large-scale rain;
LS_SNOW: large-scale snow; LS_GRPL: large-scale graupel; CV_CLIQ: convective cloud
water; CV_CICE: convective cloud ice; CV_RAIN: convective rain; CV_SNOW: convective
snow). c) the distribution of large-scale and convective cloud among the sub-columns generated
by the SCOPS scheme (i.e., frac_out from scops.f). d) the distribution of large-scale and
convective precipitation among the sub-columns generated by the SCOPS-PREC scheme (i.e.,
prec_frac from prec_scops.f). e) the mixing ratio (left panels) and effective radius (right panels)
of three precipitation hydrometeor types among the sub-columns.






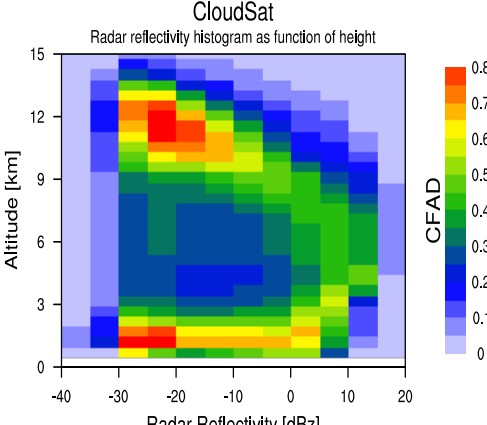

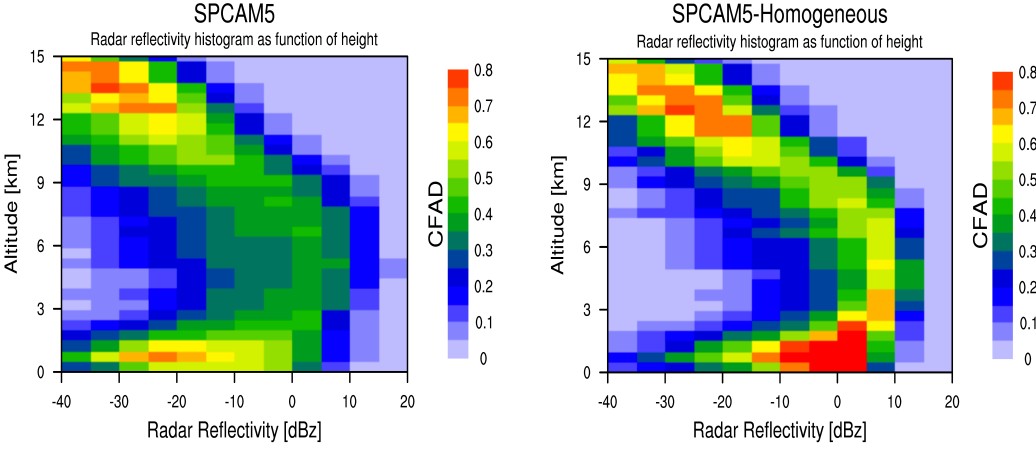



Figure 3. Tropical averaged radar reflectivity-height histogram in the CloudSat observation (top),
the SPCAM5 COSP simulation (bottom left) and the SPCAM5-Homogeneous COSP simulation
(bottom right).








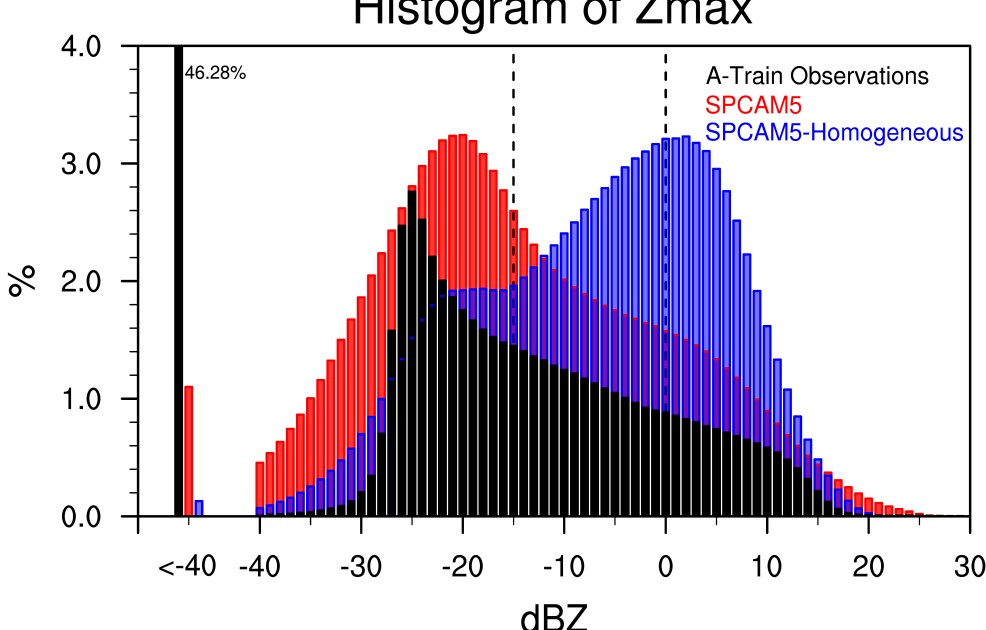



Figure 4. The histograms of column maximum radar reflectivity for liquid clouds over oceanic

regions from 45°S to 45°N in A-Train satellite observations, SPCAM5 COSP and SPCAM5-

Homogeneous COSP simulations.













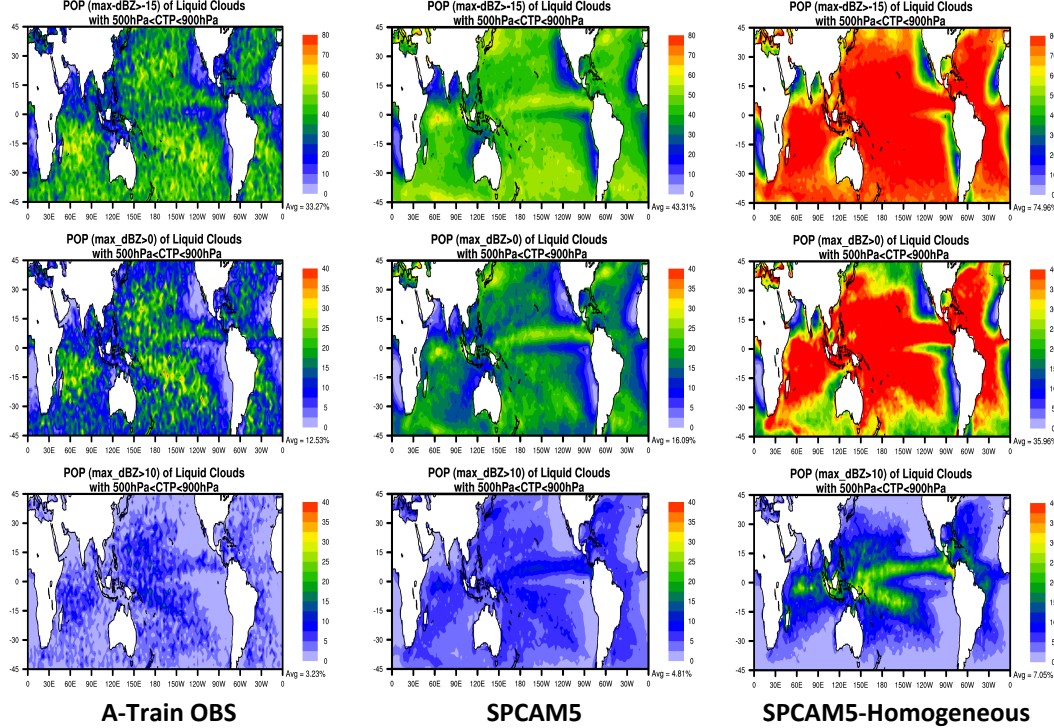

**A-Train OBS**          **SPCAM5**          **SPCAM5-Homogeneous**

Figure 5. Probability of precipitation (POP) of liquid clouds between 500hPa and 900hPa levels
in the satellite observations (left panel), the SPCAM5 COSP simulation (middle panel) and the
SPCAM5-Homogeneous COSP simulation (right panel). Three categories of precipitation:
drizzle plus rain (column $Z_{max}$ > -15 dBZ, top panels), rain (column $Z_{max}$ > 0 dBZ, middle
panels), and strong rain only (column $Z_{max}$ > 10 dBZ , bottom panels). Unit of POP is %.







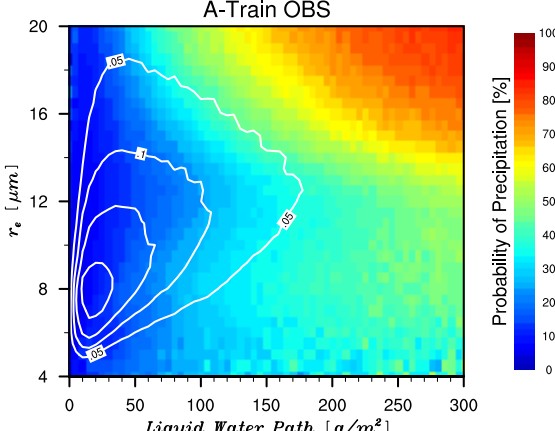

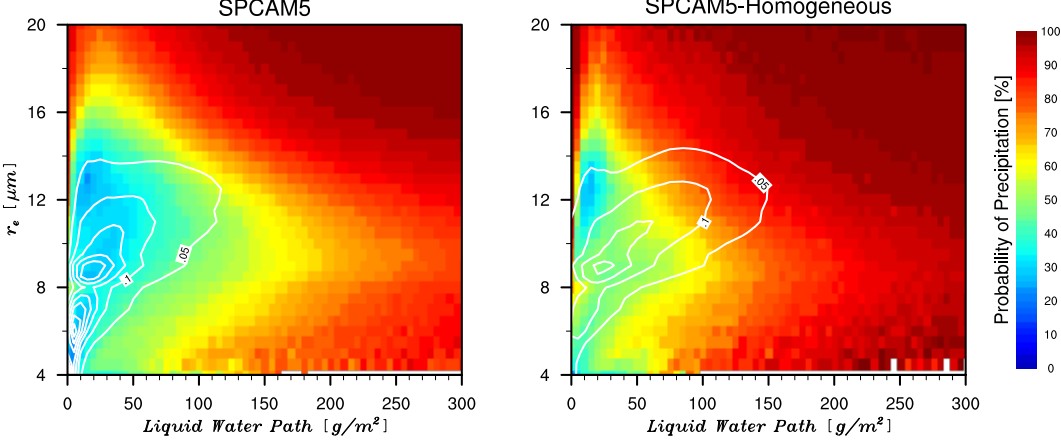



Figure 6. POP (drizzle or rain) of liquid clouds at each LWP and liquid cloud effective radius in

the satellite observations (top), the SPCAM5 COSP simulation (bottom left) and the SPCAM5-

Homogeneous COSP simulation (bottom right). The white solid contours are joint PDF of LWP

and liquid cloud effective radius. Units of POP and PDF are %.