# Peer review of "The Importance of Considering Sub-grid Cloud Variability When Using Satellite Observations to Evaluate the Cloud and Precipitation Simulations in Climate Models"

_Geoscientific Model Development, 2018_

## Referee Comment (RC1) · Anonymous Referee #1 · 12 Feb 2018

Reviewer comments: gmd-2018-13

The authors explore the sub-grid variability assumed in COSP, which many studies use to compare observations to models. Use of SPCAM at 4km resolution allows the authors to examine the impact of resolving sub-grid variability on COSP.

I really like this paper and think it is very important to get it out there to allow people to better understand the abilities of COSP and that it shouldn't be applied fecklessly to any given model. Frequently COSP is used in studies as some sort of magical talisman that

bridges models and observations. This is rarely questioned as far as I can tell. As the authors point out in line 91 page 4, there are some basic resolution issues in coupling a GCM to COSP and trying to pull out something like a satellite pixel. I would almost suggest that the authors move their comments on line 91-98 into the abstract somehow so that people who just skim it will have this brought to their attention as it is critically important. However, this change is not required scientifically and may be disregarded by the authors. This paper will be a very useful reference in the COSP documentation for people trying to set their model up to run with COSP.

Line 126- convectional=convective

Line 129- it is worth noting that this is still in the so-called convective grey zone, for example: Field et al. (2017). Do you think your results would change much if you doubled your grid size?

Line 186 'sub-columns are'

Line 262- Although not required, the authors might consider how this might contextualize results such as Nam et al. (2012).

Line 374- The authors have focused on the warm rain process representation. This may be a very ignorant comment on my part, but I would be interested in how the evaluation of the first indirect effect in GCMs might be affected by the assumptions in homogeneous COSP. For example, most empirical studies of the first indirect effect utilize level 3 gridded data (McCoy et al., 2017a;Gryspeerdt et al., 2017;Bellouin et al., 2013;Quaas et al., 2008;Quaas et al., 2009), either using observed AOD/AI (Gryspeerdt et al., 2017) or reanalysis aerosol mass (McCoy et al., 2017a;McCoy et al., 2017b). These studies compare to level 3 aggregated cloud and aerosol from models and make statements regarding the ability of models to represent the first indirect effect. If the authors could comment on whether this is a valid approach that would be highly informative.
Figure 2 c-d are somewhat hard to parse.

References Bellouin, N., Quaas, J., Morcrette, J. J., and Boucher, O.: Estimates of aerosol radiative forcing from the MACC re-analysis, Atmos. Chem. Phys., 13, 2045-2062, 10.5194/acp-13-2045-2013, 2013. Field, P. R., BrozlEková, R., Chen, M., Dudhia, J., Lac, C., Hara, T., Honnert, R., Olson, J., Siebesma, P., de Roode, S., Tomassini, L., Hill, A., and McTaggart-Cowan, R.: Exploring the convective grey zone with regional simulations of a cold air outbreak. Quarterly Journal of the Royal Meteorological Society, 143, 2537-2555, 10.1002/qj.3105, 2017. Gryspeerdt, E., Quaas, J., Ferrachat, S., Gettelman, A., Ghan, S., Lohmann, U., Morrison, H., Neubauer, D., Partridge, D. G., Stier, P., Takemura, T., Wang, H., Wang, M., and Zhang, K.: Constraining the instantaneous aerosol influence on cloud albedo, Proceedings of the National Academy of Sciences, 114, 4899-4904, 10.1073/pnas.1617765114, 2017. McCoy, D. T., Bender, F. A. M., Grosvenor, D. P., Mohrmann, J. K., Hartmann, D. L., Wood, R., and Field, P. R.: Predicting decadal trends in cloud droplet number concentration using reanalysis and satellite data, Atmos. Chem. Phys. Discuss., 2017, 1-21, 10.5194/acp-2017-811, 2017a. McCoy, D. T., Bender, F. A. M., Mohrmann, J. K. C., Hartmann, D. L., Wood, R., and Grosvenor, D. P.: The global aerosol-cloud first indirect effect estimated using MODIS, MERRA, and AeroCom, Journal of Geophysical Research: Atmospheres, 122, 1779-1796, 10.1002/2016jd026141, 2017b. Nam, C., Bony, S., Dufresne, J. L., and Chepfer, H.: The 'too few, too bright' tropical low-cloud problem in CMIP5 models, Geophys. Res. Lett., 39, n/a-n/a, 10.1029/2012GL053421, 2012. Quaas, J., Boucher, O., Bellouin, N., and Kinne, S.: Satellite-based estimate of the direct and indirect aerosol climate forcing, Journal of Geophysical Research: Atmospheres, 113, n/a-n/a, 10.1029/2007JD008962, 2008. Quaas, J., Ming, Y., Menon, S., Takemura, T., Wang, M., Penner, J. E., Gettelman, A., Lohmann, U., Bellouin, N., Boucher, O., Sayer, A. M., Thomas, G. E., McComiskey, A., Feingold, G., Hoose, C., Kristjansson, J. E., Liu, X., Balkanski, Y., Donner, L. J., Ginoux, P. A., Stier, P., Grandey, B., Feichter, J., Sedney, I., Bauer, S. E., Koch, D., Grainger, R. G., Kirkevag, A., Iversen, T., Seland, O., Easter, R., Ghan, S. J., Rasch, P. J., Morrison, H., Lamargue, J. F., Iacono, M. J., Kinne,

**GMDD**
S., and Schulz, M.: Aerosol indirect effects - general circulation model intercomparison and evaluation with satellite data, Atmospheric Chemistry and Physics, 9, 8697-8717, 2009.

---

## Referee Comment (RC2) · Anonymous Referee #2 · 6 Mar 2018

General Comments:

This is a well written paper that clearly demonstrates the importance of considering the sub-grid variability of cloud and precipitation when applying the COSP MODIS and CLOUDSAT satellite simulators. The authors demonstrate that the radar reflectivities derived from the sub-grid CRM cloud and precipitation properties, versus the grid mean properties, are vastly different and excluding sub-grid variations can lead to misinterpretation of model performance (leading to the conclusion that the drizzle or rain is triggered too frequently).

I find this work to be important as its results will impact the analysis of CMIP6 model simulations, many of which will very likely be using the oversimplified COSP sub-column generator in version 1.4.

Specific Comments:

Line 83: What is the pixel resolution of MODIS?

Line 129: A more detailed description regarding clouds and micro-physics in SPCAM would be appreciated. How can microphysical processes be resolved at 4km? Does SPCAM use the Morrison and Gettelman (2008) microphysical scheme mentioned?

Fig 2 (& related Caption) - Add experiment name to plot and caption. In regards to Subplot e) Add title to columns (ie mixing ratio / eff. radius). (FYI - I like that the authors added the variable and routine 'frac_out from scops.f' to the caption. This will be very helpful for other modelers).

Line 218: Consider sharing the modification to COSP to the community.

Line 274-247: The obs. pdf needs to be further analyzed. Finding that CloudSat only detects 54% of collocated warm clouds MODIS detects is a significant problem that needs to understood/explained further. Are you saying that a large chunk of the 46% of undetected clouds are too thin and can explain the sharp decline in the pdf around -40 to -25dBZ? If so, how often are warm liquid clouds too thin to be detected by CloudSat (check with CALIPSO)? Ground clutter really only influences the lowest approx. 1∼km. This would imply that nearly half (or some significant fraction) of the clouds MODIS detects are within the lowest 1∼km (again, check with CALIPSO). Also, is there a way of checking for frequency of attenuation (for a given altitude) in the Observations? While I understand this will very likely not change the results of this plot, it is important to note which types of clouds are being eliminated in the observations.

Line 339 / Section 4: Can you state which other COSP simulators, and how a few selected variables, would be influenced by the sub-grid cloud variability (and in-cloud mi-

Interactive
comment

crophysical properties)? Otherwise, I recommend changing broad statements of about the COSP simulator to more specific statements regarding the CloudSat simulator.

Section 4: It needs to be emphasized that the 'sub-grid variability of mass and microphysics within each hydrometeor type' is key.

Double check references.

---

## Author Comment (AC1) · 3 May 2018

The authors explore the sub-grid variability assumed in COSP, which many studies use to compare observations to models. Use of SPCAM at 4km resolution allows the authors to examine the impact of resolving sub-grid variability on COSP.

I really like this paper and think it is very important to get it out there to allow people to better understand the abilities of COSP and that it shouldn't be applied fecklessly to any given model. Frequently COSP is used in studies as some sort of magical talisman that

bridges models and observations. This is rarely questioned as far as I can tell. As the authors point out in line 91 page 4, there are some basic resolution issues in coupling a GCM to COSP and trying to pull out something like a satellite pixel. I would almost suggest that the authors move their comments on line 91-98 into the abstract somehow so that people who just skim it will have this brought to their attention as it is critically important. However, this change is not required scientifically and may be disregarded by the authors. This paper will be a very useful reference in the COSP documentation for people trying to set their model up to run with COSP.

**Thank you very much for the encouraging remarks. In our revision, we have revised our manuscript based on your helpful advices.**

Line 126- convectional=convective **Ans**: This correction is done.

Line 129- it is worth noting that this is still in the so-called convective grey zone, for example: Field et al. (2017). Do you think your results would change much if you doubled your grid size?

**Ans**: Yes, 4-km resolution is still in the so-called convective grey zone. As mentioned in Field et al. (2017), it is common practice for models operating in the convective Grey Zone to simply switch off the convection parameterization somewhere in the resolution ranging between 500 and 5km. No, we don't think our results would change much if we doubled the grid size.

Line 186 'sub-columns are' **Ans**: This correction is done.

Line 262- Although not required, the authors might consider how this might contextualize results such as Nam et al. (2012).

**Ans**: We have added a sentence to contextualize the results from previous studies such as Nam and Quaas (2012) in our revised manuscript.

Line 374- The authors have focused on the warm rain process representation. This may be a very ignorant comment on my part, but I would be interested in how the

evaluation of the first indirect effect in GCMs might be affected by the assumptions in homogeneous COSP. For example, most empirical studies of the first indirect effect utilize level 3 gridded data (McCoy et al., 2017a;Gryspeerdt et al., 2017;Bellouin et al., 2013;Quaas et al., 2008;Quaas et al., 2009), either using observed AOD/AI (Gryspeerdt et al., 2017) or reanalysis aerosol mass (McCoy et al., 2017a;McCoy et al., 2017b). These studies compare to level 3 aggregated cloud and aerosol from models and make statements regarding the ability of models to represent the first indirect effect. If the authors could comment on whether this is a valid approach that would be highly informative.

**Ans**: We have compared the simulated total cloud fraction by the MODIS, CALIPSO and CloudSat simulators, and the in-cloud properties by the MODIS simulator for the SPCAM5 and SPCAM5-Homogeneous simulations. As shown in the below figure (Figure S1), all the simulated cloud properties are influenced by the sub-grid cloud variability but to different extents. The CloudSat simulation is affected most notably since the calculation of radar reflectivity is strongly sensitive to the inhomogeneous distribution of cloud droplet size. To what extent these differences will influence the aerosol-indirect effect evaluation is beyond the scope of our study, but it'd be wise to keep in mind this potential uncertainty.

Figures 2 c-d are somewhat hard to parse.

**Ans**: Figure 2c shows the distribution of large-scale (red plus signs for $\mathrm{frac}_out = 1) and convective (blue plus signs for frac_out = 2) cloud among the sub-columns generated by the SCOPS scheme, the variable frac_out is produced in the scops.f routine. The sub-column at certain vertical level is stratiform cloudy if frac_out = 1, or convective cloudy if frac_out = 2 at that vertical level. Figure 2d shows the distribution of large-scale (red plus signs for prec_frac = 1), convective (blue plus signs for prec_frac = 2), and mixed (green plus signs for prec_frac = 3) precipitation among the sub-columns generated by the SCOPS-PREC scheme (i.e., prec_frac from prec_scops.f). We have added more detailed captions and ex$

$Please also note the supplement to this comment:$
$https://www.geosci-model-dev-discuss.net/gmd-2018-13/gmd-2018-13-AC1-supplement.pdf$

————————————————

**Total Cloud Fraction [%]**

[Figure]

**In-Cloud Properties of Liquid Cloud**

---

## Author Comment (AC2) · 3 May 2018

**General Comments:**

This is a well-written paper that clearly demonstrates the importance of considering the sub-grid variability of cloud and precipitation when applying the COSP MODIS and CLOUDSAT satellite simulators. The authors demonstrate that the radar reflectivities derived from the sub-grid CRM cloud and precipitation properties, versus the grid mean properties, are vastly different and excluding sub-grid variations can lead to misinterpretation of model performance (leading to the conclusion that the drizzle or rain is triggered too frequently).

I find this work to be important as its results will impact the analysis of CMIP6 model simulations, many of which will very likely be using the oversimplified COSP subcolumn generator in version 1.4.

**Thank you very much for the encouraging comments. We have revised our manuscript based on your constructive advices.**

**Specific Comments:**

**Line 83: What is the pixel resolution of MODIS?**

Ans: The MODIS data we used in this study is the C6 Aqua MODIS products that include the 1km geolocation products and the cloud mask product (Ackerman et al., 1998). As mentioned in Section 2.3 of our manuscript, we collocated 5 years (2006 ~ 2010) of pixel-level (i.e., level-2) MODIS and CloudSat observations using the collocation scheme developed in Cho et al. (2008). We aggregated these CloudSat and MODIS collocated level-2 data to the level-3 (gridded) data with the horizontal resolution as that in our CAM5.3, which is 1.9° latitude × 2.5° longitude.

Line 129: A more detailed description regarding clouds and microphysics in SPCAM would be appreciated. How can microphysical processes be resolved at 4km? Does SPCAM use the Morrison and Gettelman (2008) microphysical scheme mentioned?

Ans: As suggested, we have added a short paragraph to describe the physical parameterizations in SPCAM. SPCAM uses the two-moment cloud microphysics scheme of Morrison et al. (2005) to resolve microphysical processes at 4km. The Morrison and Gettelman (2008) microphysical scheme is based loosely on the approach of Morrison et al. (2005).

Fig 2 (& related Caption) - Add experiment name to plot and caption. In regards to

Subplot e) Add title to columns (ie mixing ratio / eff. radius). (FYI - I like that the authors added the variable and routine 'fracout from scops.f' to the caption. This will be very helpful for other modelers).

**Ans: We have modified Figure 2 as suggested in our revised manuscript.**

Line 218: Consider sharing the modification to COSP to the community.

Ans: The latest version of COSP (v2.0) might have already implemented the capability for sub-column sampling. But yes, we will share our finding with the COSP to the community (through personal communication and COSP user google group https://groups.google.com/forum/#!forum/cosp-user).

Line 274-247: The obs. pdf needs to be further analyzed. Finding that CloudSat only detects 54% of collocated warm clouds MODIS detects is a significant problem that needs to understood/explained further. Are you saying that a large chunk of the 46% of undetected clouds are too thin and can explain the sharp decline in the pdf around -40 to -25dBZ? If so, how often are warm liquid clouds too thin to be detected by CloudSat (check with CALIPSO)? Ground clutter really only influences the lowest approx. 1\_km. This would imply that nearly half (or some significant fraction) of the clouds MODIS detects are within the lowest 1\_km (again, check with CALIPSO). Also, is there a way of checking for frequency of attenuation (for a given altitude) in the Observations? While I understand this will very likely not change the results of this plot, it is important to note which types of clouds are being eliminated in the observations.

**Ans:**

Yes, using only the CloudSat cloud mask alone (i.e., 2B-GEOPROF product) would miss significant amount of liquid-phase clouds. In addition to surface cluttering problem, some clouds are either too thin or their particle sizes are too small to generate detectable radar echo (i.e., >-30dBz), and therefore would be missed by CloudSat. Though it should be kept in mind that CloudSat is designed to detect "hydrometer" which include both cloud and more importantly precipitation. Moreover, as you pointed out, CloudSat is flying side by side with CALIPSO which is much more sensitive to thin clouds. That is why the CloudSat team developed the 2B-GEOPROF-LIDAR product which combined the CALIPSO and CloudSat for cloud detection. In our study, we mainly use CloudSat to detect drizzle and use MODIS to detect clouds.

We could not find a published reference to quantify and explain the clouds missed by CloudSat (maybe because it is well known?), but we found two papers. one by Takahashi et al. (2017) who used CloudSat only cloud mask and the other by Kay et al. (2012) who used ISCCP, MISR and CALIPSO cloud masks. Below are the cloud fractions from the two study. It is evident that the CloudSat only cloud mask detects significantly lower cloud fraction than CALIPSO or the other two passive sensors. In particular, over the stratocumulus cloud regions (e.g., SE pacific off coast Peru and NE pacific off coast of California) the cloud fraction based on CloudSat alone is only around 50% much lower than the CALIPSO values ~ 75%~85%.

**Takahashi et al. (2017) cloudSat cloud mask**

One more point to note is that many studies have shown that the MODIS cloud mask agrees well with CALIPSO cloud mask. In fact, in our early paper, Song et al. (2018), we found that the total cloud fraction from MODIS is about 61% between 45S and 45N, only 2% lower than the CALIPSO cloud fraction. See Figure below.